# When the COVID-19 Pandemic Surges during Influenza Season: Lessons Learnt from the Sentinel Laboratory-Based Surveillance of Influenza-Like Illness in Lombardy during the 2019–2020 Season

**DOI:** 10.3390/v13040695

**Published:** 2021-04-16

**Authors:** Cristina Galli, Laura Pellegrinelli, Laura Bubba, Valeria Primache, Giovanni Anselmi, Serena Delbue, Lucia Signorini, Sandro Binda, Danilo Cereda, Maria Gramegna, Elena Pariani

**Affiliations:** 1Department of Biomedical Sciences for Health, University of Milan, 20133 Milan, Italy; cristina.galli@unimi.it (C.G.); laura.pellegrinelli@unimi.it (L.P.); laura.bubba@unimi.it (L.B.); valeria.primache@unimi.it (V.P.); giovanni.anselmi@unimi.it (G.A.); sandro.binda@unimi.it (S.B.); 2Department of Biomedical, Surgical and Dental Sciences, University of Milan, 20133 Milan, Italy; serena.delbue@unimi.it (S.D.); lucia.signorini@unimi.it (L.S.); 3DG Welfare, UO Prevenzione, Lombardy Region, 20124 Milan, Italy; danilo_cereda@regione.lombardia.it (D.C.); Maria_Gramegna@regione.lombardia.it (M.G.); 4Interuniversity Research Center on Influenza and Other Transmissible Infections (CIRI-IT), 16132 Genoa, Italy

**Keywords:** SARS-CoV-2, influenza viruses, influenza-like illness, surveillance, molecular detection, molecular epidemiology

## Abstract

This paper outlines the role of Lombardy’s regional influenza reference laboratory (Northern Italy) in the surveillance of influenza-like illnesses (ILIs) in monitoring SARS-CoV-2 circulation by analyzing 631 consecutive nasopharyngeal swabs (NPSs) collected from ILI outpatients by sentinel physicians during the 2019–2020 season. The samples were tested by specific real-time RT-PCRs targeting SARS-CoV-2, influenza viruses, and RSVs. Results: Of these NPSs, 31% tested positive for influenza viruses, 10% for SARS-CoV-2, and 7% for RSV. No coinfections were detected. Influenza viruses and RSVs circulated throughout the surveillance period until the end of February (week 9-2020), when they suddenly ceased to circulate seven weeks earlier than during the previous five influenza seasons. After the first detection of SARS-CoV-2 in our ILI outpatients at the beginning of March (week 10-2020), SARS-CoV-2 remained the only virus identified throughout the surveillance period. Patients ≥ 65 years had a 3.2-fold greater risk of being infected with SARS-CoV-2, while school-age children (5–14 years) and children < 5 years proved to be the age groups most at risk of contracting influenza viruses and RSV, respectively. Our experience demonstrates that laboratory-based ILI surveillance networks are essential for identifying SARS-CoV-2 cases that would otherwise remain undetected, in order to stop their spread within our communities.

## 1. Introduction

Since the first cases of severe acute respiratory syndrome were identified in China at the end of December 2019 [1], a pneumonia outbreak, caused by human-to-human transmission of a new coronavirus, has rapidly spread worldwide and has become a global pandemic [2,3]. On 10 February 2020, the World Health Organization (WHO) announced the spread of this novel coronavirus named SARS-CoV-2, and the disease that it causes COVID-19 [4].

In Europe, SARS-CoV-2 started to circulate simultaneously with other types of seasonal influenza, which are routinely monitored by active sentinel influenza-like illness (ILI) surveillance systems, which some European countries have had to suspend so that virological laboratories could focus on COVID-19 laboratory diagnoses [5,6].

In Italy, the first recorded community-acquired case of COVID-19 was reported in Lombardy on 20 February 2020, and a total of 75,732 COVID-19 cases were identified in this region from then until the end of April 2020 [7], leaving virological laboratories to deal with a crushing workload. As the regional reference laboratory for influenza and other respiratory viruses in Lombardy, we collaborate closely with the national COVID-19 surveillance system coordinated by the National Institute of Health (Istituto Superiore di Sanità, ISS), and soon after the global spread of SARS-CoV-2 was announced, we implemented SARS-CoV-2 molecular detection as part of the ILI sentinel surveillance program [8]. In fact, since ILI clinical manifestations are similar to mild symptoms of SARS-CoV-2 infection, the Italian Influenza Surveillance Network (InfluNet) was used to monitor the epidemiological and virological characteristics of SARS-CoV-2, which had previously been employed successfully for respiratory viruses other than influenza, such as respiratory syncytial virus (RSV) [9], enteroviruses, and parechoviruses [10]. Recently, it has been demonstrated that ILI outpatient surveillance data could be used to estimate the baseline prevalence of SARS-CoV-2 and to identify surges of excess ILIs correlated with SARS-CoV-2, with the broader potential to use this syndromic surveillance for early detection and molecular–epidemiological characterization [11,12].

This paper describes: (i) the molecular epidemiology of SARS-CoV-2, influenza viruses, and RSV during the 2019–2020 winter season in Lombardy (a Northern Italian region with nearly 10 million inhabitants, out of a total national population of 60 million), which was the Italian region most affected by COVID-19 during the first wave of the pandemic, and (ii) the use of ILI surveillance as a tool for monitoring SARS-CoV-2 circulation.

## 2. Materials and Methods

### 2.1. Influenza Surveillance Network, ILI Case Definition

InfluNet is coordinated by the Italian Ministry of Health and is based on the voluntary participation of sentinel physicians (SPs), pediatricians and general practitioners who provide medical care to the general population in ambulatory medical care facilities for ILI occurrence [7]. Weekly, the SPs report data stating the number of ILI cases observed, and they collect respiratory samples (nasopharyngeal swabs, NPSs) from a subset of their outpatients for virological surveillance. 

The standard case definition for ILIs is: sudden onset of fever (>38 °C) or feverishness, one or more respiratory symptoms (cough, sore throat, and/or shortness of breath), and one or more systemic symptoms (myalgia, headache, and/or malaise) [13]. For each ILI case, information on age, sex, influenza vaccination status, and presence of underlying medical conditions (i.e., cardiovascular diseases, chronic respiratory diseases, metabolic diseases, and immunosuppression) is collected anonymously [8]. 

At the beginning of the 2019–2020 influenza season (week 42-2019), 165 SPs were recruited for ILI epidemiological surveillance, thus covering a total of 220,069 people, which corresponds to 2.2% of the total population of Lombardy. Virological surveillance of influenza started on week 46-2019 (mid-November) and ended on week 17-2020 (the end of April) [8].

### 2.2. Study Population and Study Design

All NPSs collected for virological surveillance from ILI cases were tested on a weekly basis for influenza and RSV. SARS-CoV-2 molecular detection was implemented by week 7-2020 (10 February, when the WHO launched the global alert), and after that date, all NPSs were tested daily for SARS-CoV-2. All respiratory samples collected from week 46-2019 (11 November 2019) to week 6-2020 (9 February 2020) were retrospectively tested for SARS-CoV-2.

Epidemiological and virological characteristics of ILI cases were described by week and by age group (0–4 years, 5–14 years, 15–64 years, ≥65 years), comparing the data of the 2019–2020 season with those of the five previous influenza seasons (from 2014–2015 to 2018–2019) in Lombardy.

### 2.3. Molecular Detection of Influenza Viruses, RSV, and SARS-CoV-2

NPSs were collected with flocked swabs in a universal transport medium (Sigma Virocult^®^ kit, Medical Wire—MWE, Corsham, Wiltshire, UK). RNA was extracted by the commercial method QIAamp Viral RNA Mini kit (QIAGEN GmbH, Hilden, Germany), following the manufacturer’s instructions. In order to check extraction performance, a one-step, real-time RT-PCR was carried out, targeting the human ribonuclease P gene (RNase P); the samples showing cycle threshold (Ct) values of <35 were considered suitable to be tested for molecular virological analyses [14]. All specimens were analyzed using a specific, one-step RT-PCR to simultaneously detect influenza viruses of types A and B using specific primer/probe sets targeting the matrix (M) and the nucleoprotein (NP) genes, respectively, in accordance with the US Centers for Disease Control and Prevention’s (CDC) protocols for influenza surveillance [15]. Influenza A virus-positive samples were further subtyped by a one-step RT-PCR assay by using specific primer/probe sets for the hemagglutinin (HA) gene to discriminate between A(H1N1)pdm09 and A(H3N2) subtypes. Influenza B virus-positive specimens were tested by a multiplex, one-step RT-PCR assay targeting the HA gene to discriminate the B/Yamagata and B/Victoria lineage [14,15]. A one-step, real-time RT-PCR assay was performed to detect RSV by using a specific primer/probe set, targeting the matrix (M) gene [16]. SARS-CoV-2 RNA was detected by a one-step, real-time RT-PCR assay, amplifying different portions of the nucleocapsid (N) gene in accordance with CDC protocol [17]. A sample was considered positive for the specific viral target when its Ct value was <40.

### 2.4. Statistical Analysis

Statistical analysis was performed using the Open Source Epidemiologic Statistics for Public Health OpenEpi, version 3.03 [18]. The frequency of positive samples was expressed as a crude proportion, with the corresponding 95% confidence interval (95% CI) calculated by the Mid-P exact test, assuming a normal distribution. The inter-quartile range (IQR) was computed as the difference between the first and third quartiles of the age distribution. The positivity rate from each virus was calculated as the number of laboratory-confirmed infections out of the total number of individuals with ILIs, plus a specific characteristic. Proportions between groups were compared using the Mid-P exact test based on binomial distribution. For continuous variables, the paired t-test was used. A *p*-value < 0.05 was considered significant (two-tailed test). The odds ratio (OR) was computed. ILI incidence (per 1000 inhabitants) by week, as recorded in the 2019–2020 season in Lombardy, was obtained from the InfluNet database [8].

## 3. Results

As shown in Table 1, a total of 631 NPSs were collected from 631 individuals during the 2019–2020 influenza season (from week 46-2019 to week 17-2020). The median age was 32 years (IQR: 42 years), 53% were aged 15 to 64 years, 21% were aged 5 to 14 years, and children < 5 years and the elderly ≥ 65 years accounted for 16% and 10% of the study population, respectively. Overall, 311 (49%) of the ILI cases were males, with no statistical gender differences (*p* = 0.4). In total, 24% of the ILI cases had at least one pre-existing, underlying medical condition.

Overall, 31% (*n* = 197) of ILI cases tested positive for influenza virus, 7% (*n* = 46) for RSV, and 10% (*n* = 66) for SARS-CoV-2; no coinfections were detected.

Figure 1 shows the temporal distribution of ILI cases and virus detection according to the week of sample collection. Influenza viruses and RSVs were detected from the beginning of the surveillance period up to week 9-2020, when SARS-CoV-2, which was detected for the first time in week 10-2020, prevailed over them and then remained the only virus detected until the end of surveillance (week 17-2020). Influenza viruses and RSVs disappeared seven weeks earlier than in the previous five influenza seasons (Figure 2) and did not coincide with a decrease in ILI incidence, which rose again from week 10-2020 (Figure 1a).

During the 2019–2020 season, influenza viruses were detected from week 48-2019 until week 10-2020, with a positivity rate higher than 50% from week 5-2020 to week 9-2020, and a peak (positivity rate = 64.4%) in week 7-2020. It is noteworthy that in week 11-2020, ILI incidence was more than two times greater than that observed in the same week of the previous five seasons, when ILI incidence was nearly three-fold lower (6‰ vs. 2.4‰ median incidence from 2014–2015 to 2018–2019) (Figure 3), and influenza viruses were detected up to week 17 (Figure 2a). The same pattern was observed for RSV during the 2019–2020 season, which was first detected in week 46-2019, peaked in week 6-2020 when 12.5% of samples tested RSV-positive, and disappeared in week 9-2020. This is in contrast with the trend observed from the 2014–2015 influenza season onwards, when RSVs were detected up to week 13 (Figure 2b).

In our ILI series, SARS-CoV-2 was detected for the first time in week 10-2020 and was the only respiratory virus detected thereafter; the highest positivity rate was observed in week 11-2020, when 71% of ILI cases tested positive for SARS-CoV-2. It is noteworthy that the retrospective analysis of our ILI samples collected from week 46-2019 to week 9-2020 revealed that none of the samples tested positive for SARS-CoV-2 RNA.

Table 1 summarizes the characteristics of the ILI cases that tested positive for influenza viruses, RSVs, and SARS-CoV-2. In our ILI series, sex distribution was similar for influenza virus-positive cases (males vs. females: 48% vs. 49%; *p* = 0.807) and RSV-positive (males vs. females: 52% vs. 49%; *p* = 0.695), while SARS-CoV-2 was detected more frequently among males than females (61% vs. 39%, *p* < 0.0001). Data analysis by age group showed that influenza viruses and RSVs were detected in all age groups, while children < 15 years proved to be those most affected by these viruses. In particular, influenza positivity rates were highest (63%) among school-age children (5–14 years), with a nearly six-fold higher risk of infection (OR: 5.8; 95%CI: 3.9–8.9) than the other age groups, while children < 5 years were those most affected by RSVs, with a 31% positivity rate and a nearly 16-fold higher risk of infection (OR: 15.6; 95%CI: 8.1-31.1) than the other age groups (Table 2).

Contrastingly, in our ILI series, no SARS-CoV-2-positive cases were observed among children aged 0–4 years, and only 1.5% of SARS-CoV-2-positive outpatients were aged 5–14 years. The median age of SARS-CoV-2-positive cases was 53 years (IQR: 39.5 years; range: 14–90 years), which was significantly older (*p* < 0.001) than the median age of influenza virus-positive cases (median age: 23 years; IQR: 32 years; range: 0–82 years) or RSV-positive cases (median age: 15 years; IQR: 23.2 years; range: 0–76 years). This difference in age distribution among SARS-CoV-2-positive cases was also due to the fact that individuals belonging to the 15–64 age group (OR: 3.0; 95% CI: 1.7–5.5) and those aged ≥ 65 years (OR: 3.2; 95%CI: 1.6–5.9) are at greater risk of infection (Table 2). These results are also supported by the age shift in ILI incidence observed from week 10-2020: in fact, up until then, most ILI cases were reported among children < 15 years, and thereafter, most ILI cases were reported in the 15–64 and ≥65 years age groups (Figure 1). This observation was in agreement with the initial trend of SARS-CoV-2 transmission, which was detected more frequently among adults (15–64 years) and the elderly (≥65 years).

Data analysis in the presence of pre-existing, underlying medical conditions in ILI cases showed that the number of patients with comorbidities was statistically higher among SARS-CoV-2-positive outpatients than among influenza virus-positive (41% vs.16%; *p* < 0.0001) or RSV-positive (41% vs. 22%; *p* = 0.03) cases. No differences (*p* = 0.4) were observed between influenza-positive and RSV-positive cases. The risk of SARS-CoV-2 infection was nearly three-fold (OR: 2.7, 95%CI: 1.5–4.7) higher for individuals with pre-existing medical conditions.

## 4. Discussion

The aim of this study was to outline the molecular epidemiology of influenza viruses, RSV, and the effects of SARS-CoV-2 circulation on ILIs during the 2019–2020 influenza season in Lombardy, the Italian region most affected by COVID-19 pandemic during the first wave. This study was conducted within the framework of the epidemiological and virological surveillance of ILIs, which is routinely carried out during the winter season; when SARS-CoV-2 started circulating in Europe, several laboratories had to adjust their routine investigations of influenza viruses and other respiratory viruses to focus on SARS-CoV-2 response [3]. Soon after the global spread of SARS-CoV-2 was announced, as the regional reference laboratory for influenza surveillance in Lombardy, we decided to extend our virological surveillance activities to include SARS-CoV-2 detection among the ILI cases identified in the area under surveillance. This provided early evidence of the introduction and community transmission of SARS-CoV-2 by week 10-2020 among outpatients with mild respiratory symptoms and corroborated the importance of an established network as a tool for the rapid detection and assessment of novel airborne infections.

The sudden disappearance of influenza viruses and RSV circulation observed at the end of February for the 2019–2020 season was concurrent with the introduction of SARS-CoV-2, as observed in other Italian regions [6] and in other countries [19,20], where the influenza virus peaked, showing positivity levels above 50%, for only two weeks, while in previous seasons, the influenza positivity rate exceeded 50% for approximately six weeks [6]. Although the shorter influenza season may have been due to laboratory response to the COVID-19 pandemic and the implementation of non-pharmaceutical interventions (such as national lockdowns and closures of schools, restaurants, offices, and factories) [21], it could have been due to viral interference, meaning that SARS-CoV-2 may have inhibited the circulation of influenza viruses and RSVs. The sudden disappearance of influenza and RSVs was concomitant with variations in the age of ILI cases, which were mainly observed in adults and the elderly rather than in children. This is especially evident for RSVs in our series, which were mainly detected among children < 5 years: as the number of ILI cases in children aged 0–4 years decreased, no RSV-positive cases were identified. Similarly, while school-aged children (5–14 years) were those at highest risk of infection from influenza prior to the introduction of SARS-CoV-2, no ILI cases tested positive for the influenza virus after week 10-2020. Contrastingly, almost all (98%) SARS-CoV-2-positive cases were observed among adults and the elderly, who had at least a three-fold higher risk of SARS-CoV-2 infection than individuals < 15 years of age. However, this shift may be due to school closures, which may have reduced the transmission of the influenza virus and RSV circulation, as previously observed by other authors [22]. In addition, in non-closure settings, it must be taken under consideration the role that children and young people may play in spreading the SARS-CoV-2 infection. In fact, it has been proven that a younger age correlates strongly with asymptomatic and mild infections and that children can be hidden drivers [23].

Higher SARS-CoV-2 infection rates were observed in men, adults, the elderly, and those with pre-existing, underlying health conditions [5,24,25], as observed in our ILI series. In fact, influenza-, RSV-, and SARS-CoV-2-positive outpatients showed distinctive and contrasting epidemiological characteristics. The proportion of males to females was equally distributed among overall ILI cases and among those who tested influenza/RSV-positive; conversely, men were more likely to be infected with SARS-CoV-2. In addition, the median age of influenza/RSV-positive ILI cases was significantly younger than that of the SARS-CoV-2-positive cases. Underlying medical conditions were reported more often among SARS-CoV-2-positive cases who had a three-fold higher risk of SARS-CoV-2 infection than those without pre-existing medical conditions. This observation may be affected by age bias. In our ILI series, no SARS-CoV-2, influenza virus, or RSV coinfections were observed, contrary to individuals with pneumonia [26,27,28].

Our study provides evidence that virological surveillance of ILI proves to be a rapid and cost-effective way of monitoring the introduction and evolution of SARS-CoV-2, as suggested by other authors [11,12]. Furthermore, it is evident that ILI sentinel surveillance in this outpatient setting, as well as other settings (hospital-based surveillance, mortality surveillance), can provide a robust laboratory-based approach for monitoring molecular epidemiology and understanding the global risk posed by SARS-CoV-2. In fact, this network is already in place and relies on skilled practitioners with experience in identifying suspected cases and managing the collection of respiratory samples. Moreover, the network follows the guidelines laid out in the pandemic preparedness and response plan in order to ensure the rapid identification and molecular characterization of new viruses, thus facilitating comparison with baseline data and providing a weekly report of new cases.

In our ILI series, no SARS-CoV-2-positive cases were identified before week 10-2020, which is in line with the findings obtained in other molecular studies carried out on Italian SARS-CoV-2 strains identified in symptomatic individuals [29,30,31]. On the other hand, ILI surveillance carried out on a greater number of individuals by a larger number of sentinel practitioners may have been able to detect SARS-CoV-2 at an earlier stage of its introduction in Italy, as the first two imported cases were detected at the end of January [32], approximately a month before the first Italian COVID-19 case was reported in Lombardy. For that reason, the guidelines established by the European Centers for Disease Control and Prevention (ECDC) strongly recommend increasing the coverage of the population under surveillance to at least 4%, instead of 2%, during the 2020–2021 season [33]. In our experience, a wider coverage of the population under surveillance would have been difficult to implement in March 2020, considering the huge number of general practitioners involved in the COVID-19 pandemic response and the impact of SARS-CoV-2 diagnoses being carried out only in a limited number of reference laboratories, which was the main limitation of our study. In fact, following the Italian lockdown (week 11-2020), a 25% drop in the number of sentinel practitioners involved in ILI surveillance was observed [34]; therefore, our SARS-CoV-2 circulation results may be underestimated.

## 5. Conclusions

Our experience demonstrated that the laboratory-based ILI surveillance network efficiently and effectively identified SARS-CoV-2 cases which would have otherwise remained undetected. In fact, syndromic surveillance enabled us to monitor these pathogens with overlapping clinical manifestations (i.e., ILIs), understand their circulation within the region and their transmissibility, and obtain baseline information that could be used for implementing successful mitigation strategies and early warning and control measures. A sentinel laboratory-based surveillance system will be able to rapidly monitor any changes in the epidemiology of a pathogen and detect the introduction of new viral variants.

## Figures and Tables

**Figure 1 viruses-13-00695-f001:**
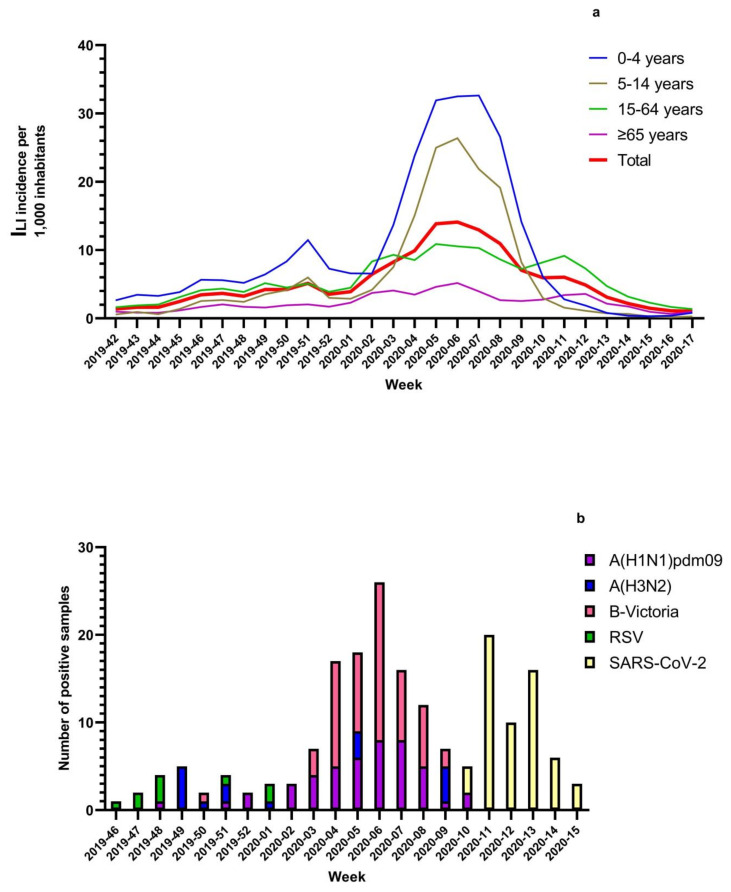
(**a**) Incidence of ILI cases (per 1000 inhabitants) by week, overall (red line), and by age group in Lombardy during the 2019–2020 season; (**b**) Number of influenza viruses- (by type/subtype), RSV-, and SARS-CoV-2-positive cases by week during the 2019–2020 season in Lombardy.

**Figure 2 viruses-13-00695-f002:**
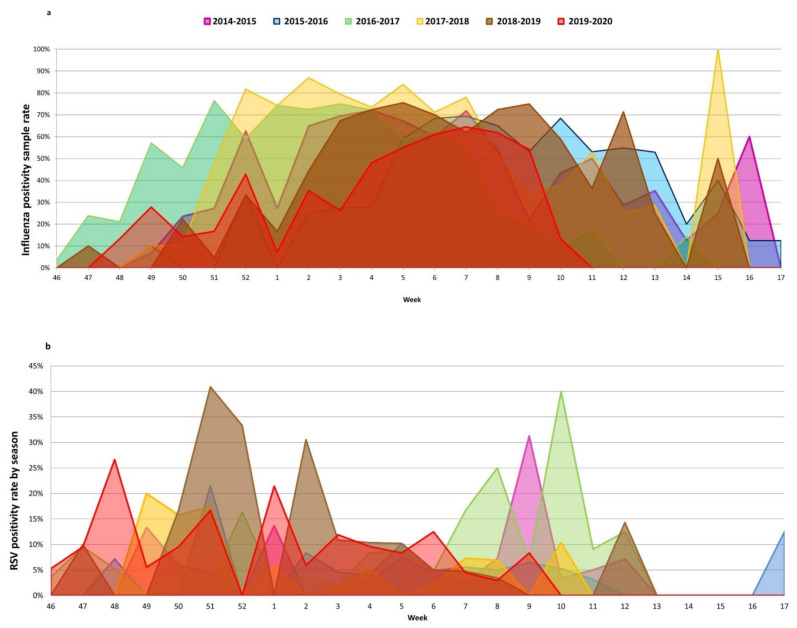
Comparison of positivity rates of influenza virus (**a**) and RSVs (**b**) by week of sample collection between the 2019–2020 season and the last five consecutive winter seasons (from 2014–2015 to 2018–2019).

**Figure 3 viruses-13-00695-f003:**
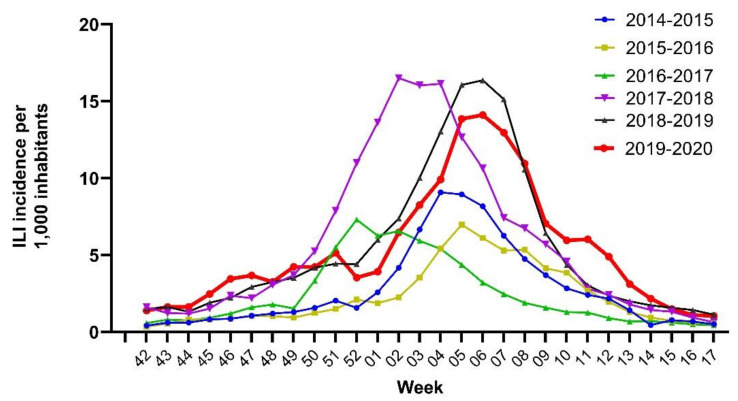
Trends of ILI incidence (per 1000 inhabitants) by week of sample collection and by winter season (from 2014–2015 to 2019–2020) in Lombardy.

**Table 1 viruses-13-00695-t001:** Total number and proportion of the main characteristics of Influenza-like illness (ILI) cases: number and positivity rates of influenza virus-positive, RSV-positive, and SARS-CoV-2-positive cases overall, and by sex, presence of underlying conditions, and age group. Median age and inter-quartile range (IQR) of all ILI patients and positive cases are also reported. Footnote: Of the 197 influenza virus-positive samples, 23% (*n* = 46) tested positive for A(H1N1)pdm09, 22% (*n* = 43) for A(H3N2), and 55% (*n* = 198) for influenza B virus, all belonging to B/Victoria lineage.

	ILI Cases	Influenza		RSV		SARS-CoV-2	
	Total Number (%)	No. Positive	Positivity Rate	No. Positive	Positivivity Rate	No. Positive	Positivity Rate
**Total**	631	197	31.2%	46	7.3%	66	10.5%
**Males**	311 (49.3%)	95	30.5%	24	7.7%	40	12.9%
**Median age**	32 years	23 years	-	15 years	-	53 years	-
**IQR** **[range]**	42 years [0–93 years]	32 years [0–82 years]	-	23.2 years [0–76 years]	-	39.5 years [14–90 years]	-
**Presence of underlying medical conditions**	149 (23.6%)	32	21.5%	10	6.7%	28	18.8%
**Age-group breakdown**							
**0–4 years**	99 (15.7%)	27	27.3%	31	31.3%	0	-
**5–14 years**	131 (20.7%)	83	63.4%	3	2.3%	1	0.8%
**15–64 years**	338 (53.6%)	79	23.4%	8	2.4%	50	14.8%
**≥65 years**	63 (10.0%)	8	12.7%	4	6.3%	15	23.8%

**Table 2 viruses-13-00695-t002:** Risk of infection from influenza viruses, RSV, and SARS-CoV-2 calculated as odds ratio (OR) by age group; 95% confidence interval (95%CI) range is also reported. Significant OR values are shown in bold.

	Risk of Infection					
Age Group	Influenza		RSV		SARS-CoV-2	
	OR	95%CI	OR	95%CI	OR	95%CI
0–4 years	0.8	0.5–1.3	**15.6**	**8.1–31.1**	/	/
5–14 years	**5.8**	**3.9–8.9**	0.2	0.1–0.7	0.1	0.01–0.3
15–64 years	0.4	0.3–0.6	0.1	0.1–0.3	**3.0**	**1.7–5.5**
>64 years	0.3	0.1–0.6	0.8	0.2–2.3	**3.2**	**1.6–5.9**

## Data Availability

The datasets generated for this study are available on request from the corresponding author.

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
