# Peer review of "When the COVID-19 Pandemic Surges during Influenza Season: Lessons Learnt from the Sentinel Laboratory-Based Surveillance of Influenza-Like Illness in Lombardy during the 2019–2020 Season"

_viruses, 2021, doi:10.3390/v13040695_

Round 1

Reviewer 1 Report

This article is based on a clinical trial in which it is used a laboratory-based network to monitor influenza-like illness to detect cases of SARS-CoV-2. The approach allows effective recognition of cases with identical symptoms. Thus, conditions caused by different pathogens but with overlapping clinical manifestations are observed. The analyzes obtained from data from the Italian Influenza Surveillance Network (InfluNet) allow the preparation of initial information for early warning and successful mitigation strategies and control measures. This laboratory approach allows rapid monitoring of any change in the epidemiology of pathogens, to quickly detect the introduction of new viral variants.

My comments on this article are:

  1. The text from lines 213 to 240 will be very convincing if it is included in the introduction to the article.
  2. The English language needs language correction.

Author Response

We thank the reviewer for his/her positive comments. We agree wholeheartedly that SARS-CoV-2 is often asymptomatic in younger individuals. We have added in the discussion section (Lines 67-71) a sentence (and related reference) to point out that.

Reviewer 2 Report

This is an interesting study about surveillance of influenza-like illness (ILI) in Lombardy during 2019-2020 season. The authors analyzed 631 consecutive nasopharyngeal swabs (NPS) collected from outpatients with ILI. 31% of samples were positive for influenza viruses, 10% for SARS-CoV-2 and 7% for RSV. Influenza viruses and RSV circulated from the beginning of surveillance activities until the end of February (week 9-2020), with a sudden stop of their spread seven weeks earlier compared to the previous five influenza seasons. The risk of infection from SARS-CoV-2 was 3.2-fold higher for patients ≥65 years, whereas the greater risk of infection from influenza virus and RSV was in school-age children (5-14 years) and in children <5 years, respectively.

The paper is very interesting and well written. However, some issues remain.

When discussing the mean age of every virus group, please take into consideration that SARS-CoV2 is often asymptomatic in younger subjects.

Author Response

  1. The text from lines 213 to 240 will be very convincing if it is included in the introduction to the article.

A: We thank the reviewer for his/her comment.

As suggested, we have amended the introduction section (lines: 292-294) by adding a sentence that emphasizes the use of ILI surveillance as a useful tool for searching for several pathogens other than influenza viruses.

  1. The English language needs language correction.

A: The manuscript has been thoughtfully revised by a native English speaker.
